# Rural–Urban Differences in the Factors Affecting Depressive Symptoms among Older Adults of Two Regions in Myanmar

**DOI:** 10.3390/ijerph18062818

**Published:** 2021-03-10

**Authors:** Yuri Sasaki, Yugo Shobugawa, Ikuma Nozaki, Daisuke Takagi, Yuiko Nagamine, Masafumi Funato, Yuki Chihara, Yuki Shirakura, Kay Thi Lwin, Poe Ei Zin, Thae Zarchi Bo, Tomofumi Sone, Hla Hla Win

**Affiliations:** 1Department of International Health and Collaboration, National Institute of Public Health, Wako City 351-0197, Japan; 2Graduate School of Medical and Dental Sciences, Niigata University, Niigata City 951-8510, Japan; yugo@med.niigata-u.ac.jp (Y.S.); nakayu0821@gmail.com (Y.C.); yshira@med.niigata-u.ac.jp (Y.S.); 3National Center for Global Health and Medicine, Bureau of International Health Cooperation, Tokyo 162-8655, Japan; i-nozaki@it.ncgm.go.jp; 4Department of Health and Social Behavior, Graduate School of Medicine, The University of Tokyo, Tokyo 113-0033, Japan; dtakagi-utokyo@umin.ac.jp; 5Department of Family Medicine, Tokyo Medical and Dental University, Tokyo 113-8510, Japan; yuiko.mail@gmail.com; 6Harvard T.H. Chan School of Public Health, Boston, MA 02115, USA; masafumifunato2@gmail.com; 7Department of Preventive and Social Medicine, University of Medicine 1, Yangon 245, Myanmar; kaythilwin.ktl2@gmail.com (K.T.L.); poeeizin1988@gmail.com (P.E.Z.); thaezarchibo@gmail.com (T.Z.B.); prof.hlahlawin@gmail.com (H.H.W.); 8Vice President, National Institute of Public Health, Wako City 351-0197, Japan; sone.t.aa@niph.go.jp

**Keywords:** depressive symptom, older adult, urban and rural area, gender differences, Myanmar

## Abstract

The aim of the study was to investigate rural–urban differences in depressive symptoms in terms of the risk factors among older adults of two regions in Myanmar to provide appropriate intervention for depression depending on local characteristics. This cross-sectional study, conducted between September and December, 2018, used a multistage sampling method to recruit participants from the two regions, for face-to-face interviews. Depressive symptoms were assessed using the 15-item version of the Geriatric Depression Scale (GDS). Depressive symptoms were positively associated with living in rural areas (B = 0.42; 95% confidence interval (CI): 0.12,0.72), female (B = 0.55; 95% CI: 0.31,0.79), illness during the preceding year (B = 0.68; 95% CI: 0.45,0.91) and non-Buddhist religion (B = 0.57; 95% CI: 0.001,1.15) and protectively associated with education to middle school level or higher (B = −0.61; 95% CI: −0.94, −0.28) and the frequency of visits to religious facilities (B = −0.20; 95% CI: −0.30, −0.10). In women in urban areas, depressive symptoms were positively associated with illness during the preceding year (B = 0.78; 95% CI: 0.36, 1.20) and protectively associated with education to middle school level or higher (B = −0.67; 95% CI: −1.23, −0.11), middle or high wealth index (B = −0.92; 95% CI: −1.59, −0.25) and the frequency of visits to religious facilities (B = −0.20; 95% CI: −0.38, −0.03). In men in rural areas, illness during the preceding year was positively associated with depressive symptoms (B = 0.87; 95% CI: 0.33, 1.42). In women in rural areas, depressive symptoms were positively associated with illness during the preceding year (B = 0.83; 95% CI: 0.36, 1.30) and protectively associated with primary education (B = −0.62; 95% CI: −1.12, −0.12) and the frequency of visits to religious facilities (B = −0.44; 95% CI: −0.68, −0.21). Religion and wealth could have different levels of association with depression between older adults in the urban and rural areas and men and women. Interventions for depression in older adults should consider regional and gender differences in the roles of religion and wealth in Myanmar.

## 1. Introduction

Depressive symptoms are a serious concern for older adults and one of the leading causes of disease burden worldwide [1]. They have been shown to predict death [2], admission to a nursing home [3] and functional decline [4]. Although depression in older adults tends to be overlooked, partly because the population exhibits atypical depressive symptoms [5], a meta-analysis of depression in older adults indicated that their depression was associated with poor health status, a new medical illness, disability, poor self-perceived health and prior depression [6]. Risk factors leading to the development of late-life depression likely comprise complex interactions among genetic vulnerabilities, cognitive diathesis, age-associated neurobiological changes and stressful events [7]. The common pathways to depression in older adults are assumed to be curtailment of daily activities and changes in social environments such as family functionality, social contact and eating status [7,8,9,10]. However, the prevalence of depression and these factors could vary substantially within countries and between countries [11,12,13]. Some of this variation could be explained by differences in methodology, subject characteristics and/or by cultural differences [5].

As the aging population grows in Asian countries, the socioeconomic cost of depression in older adults is vast, given that depressive symptoms are common [14]. In developing countries, such as Myanmar, the situation is severe, as effective medical care systems are underdeveloped [15,16]. The proportion of the population aged ≥60 years in Myanmar is anticipated to increase from about 10% in 2019 to nearly 18.5% by 2050 [17].

The pattern of Rural–urban differences in depression differs between countries. For example, the studies conducted in countries such as Britain, Canada, Finland, Japan, Korea, Taiwan, United States and Vietnam have shown either no significant differences [18,19,20,21], or rural advantages [22,23,24,25,26,27], while those conducted in China have consistently reported that older adults in rural areas displayed higher levels of depressive symptoms relative to their urban counterparts [21,28,29,30,31].

In Myanmar, religion is an essential aspect of life and central to the concept of personal identity [32]. According to a review, people with high levels of general religious involvement, organizational religious involvement, religious salience and intrinsic religious motivation were at lower risk for depressive symptoms and depressive disorders relative to other individuals [33]. In a previous study conducted in Iran [34], levels of religiosity of urban and rural residents differed significantly and the level of religiosity in rural residents tended to be higher relative to that in urban residents. A study conducted in China indicated that urban and rural residents could have different levels of individual or social motivation to practice religion and despite the hostile rural social environment, the religious residents tend to be more likely to report higher level relative to their peers in urban areas [35].

The first large-scale survey involving older adults in Myanmar revealed that, depending on the threshold applied to the short version of the Geriatric Depression Scale (GDS-4), approximately 16% to 56% of the surveyed older adults reported indications of depressive symptoms. Moreover, they found that both economic and health status were statistically significantly associated with depressive symptoms [36].

However, published papers, including those examining urban-rural differences in Myanmar, are scarce due to constraints in research and international publications during the military-controlled administration from 1962 to 2011 [37,38]. To our knowledge, no previous studies have examined how depressive symptoms differ between older adults in rural and urban areas, or whether or not their religious beliefs or practices are associated with their depressive symptoms in these areas in the Myanmar context. The current study aimed to examine whether a series of social, health and religious characteristics at the individual levels account for rural–urban differences in depressive symptoms among older adults of two regions in Myanmar.

## 2. Materials and Methods

### 2.1. Study Design and Participants

This was a cross-sectional baseline survey of a longitudinal study, “Healthy and Active Aging in Myanmar”, which examined the predictors of physical and psychological health in 1200 community-dwelling older Myanmar adults aged ≥60 years in 2018. The target population were those in the urban area of the Yangon region and the rural area of the Bago region, 91 km northeast of Yangon.

Multistage random sampling was conducted in the two regions. There are 35 townships in the Yangon region and 28 townships in the Bago region. Six townships were randomly selected from each region via population proportionated sampling. In the Yangon region, 10 wards were further randomly selected from each township. In the Bago region, 10 village tracts were randomly selected from each township. In rural areas, there are multiple villages within a single village tract. In such cases, one of the villages was randomly selected from the village tract. The differences between wards and village tracts is the degree of urbanization. The ward is the minimum unit of a residential district in an urban area and the village tract is the corresponding level in rural areas. Although wards and village tracts sometimes co-exist within a township, only wards were selected from Yangon and only village tracts were selected from Bago to capture the features of urban and rural areas from each region; we considered Yangon as representative of urban areas and Bago of rural areas. Trained surveyors visited homes of participants with the public health nurse in each community. In Yangon, the surveyors visited 1083 older adult; 610 were at home. Ten were excluded due to refusing the survey (*n* = 6) or to severe dementia or being bedridden (*n* = 4; the response rate was 98.4% in Yangon. In Bago, surveyors visited 1044 older adult; 694 were at home. Ninety-four were excluded due to severe dementia or being bedridden, thus, the response rate was 86.5% in Bago. In total, six hundred people each from the Yangon (222 men and 378 women) and Bago regions (261 men and 339 women) were surveyed.

### 2.2. Study Tools

A structured questionnaire was developed for face-to-face interviews, following the Japan Gerontological Evaluation Study (JAGES), which is a nationwide, population-based, prospective cohort study for older community-dwelling Japanese adults in Japan [39]. The linguistic translation and validation process followed the “Linguistic Validation Manual for Health Outcome Assessments [40]. It was first translated into English. Thereafter, it was translated into the local language (Burmese) and back translated into English to ensure clarity and consistency.

Research staff from the Myanmar Perfect Research Company, a group that conducts epidemiological surveys in Myanmar, were hired. The interviewers were recruited from the company. Before the commencement of the actual survey, a two-day training course on the research protocol, administration of the questionnaire and ethical concerns was conducted for the interviewers.

A pilot study was carried out before the actual survey in Urban Health Center, Dagon township, Yangon. Participants were older adults aged 60 years or older who came to the center’s out-patient clinic. We recruited 25 respondents who provided consent to participate in the pilot study, in June 2018. During the pilot study, the interviewers ensured sequence, flow and clarity of the study. After the feedback from the interviewers, the questionnaire was revised accordingly. We have determined that the above procedure ensured the face validity and content validity of the applied questionnaire.

The inclusion criteria were age of ≥60 years and residence in a selected ward or village tract. The exclusion criteria were being bed-ridden or having severe dementia. Severe dementia was defined as an Abbreviated Mental Test (AMT) score of ≤6 [41,42].

We assessed depressive symptoms using the 15-item version of the GDS (GDS-15), which was validated previously [43,44,45,46,47]. The GDS involves a simple yes/no format that is easy to administer and score [47,48]. GDS includes the following questions: (1) Are you satisfied with your current life; (2) Do you sometimes feel there is no point in living; (3) Do you think your energy for daily life or your interest in what’s going on in the world has been decreasing; (4) Do you feel your life is empty; (5) Do you often feel bored; (6) Do you usually feel good; (7) Do you feel something bad is going to happen; (8) Do you think you are fortunate; (9) Do you often feel helpless; (10) Do you prefer staying at home over going out; (11) Do you think you are more forgetful than others; (12) Do you think life is wonderful; (13) Do you feel full of energy; (14) Do you think there is no hope in your life; (15) Do you think others are better off than you are; The scores range from 0 to 15 points, with a higher score indicating a more depressive state [49].

Variables with a variance inflation factor (VIF) of >5 with other variables were excluded. The remaining variables reflecting sociodemographic characteristics were entered in a linear regression model. They included information regarding residential area (Yangon or Bago regions), sex, age, illness during the preceding year, educational level (no school, monastic, some/all primary school, middle/high school or higher), wealth index, religion (Buddhism or other), frequency of visits to religious facilities (none, a few times per year, one to three times per month, once per week, twice or three times per week, four or more times per week). The wealth index used as an economic indicator was calculated from household asset items using a method described in a previous report [50]. Principal component analysis was performed on the asset items (e.g., radio, black and white television, color television, Video/DVD player, electric fan, refrigerator, computer, store-bought furniture, personal music player, washing machine, gas cooker, electric cooker or rice cooker, air conditioner, bicycle, motorcycle, van/truck, microwave oven, mobile telephone and internet) and the principal component score was calculated based on the participants’ possession of each item and used as the wealth index.

### 2.3. Statistical Analysis

We calculated rates for each category of sociodemographic variables for the 1200 participants. Skewness and kurtosis were calculated for the distribution of GDS scores. Then, Wilcoxon rank-sum test and chi-square tests were used to compare GDS scores and socio-demographic variables between urban and rural areas. We performed linear regression analysis to examine differences in risk factors for depressive symptoms between older adults from the two types of area. GDS scores were entered as continuous variables. The multivariate adjusted results were expressed as non-standardized coefficients with 95% confidence intervals (CI). We used STATA14 (StataCorp, College Station, TX, USA) to perform all statistical analyses and the statistical significance level was set at *p* < 0.05.

### 2.4. Ethical Considerations

The survey protocol was reviewed and approved by the ethical review committee at the department of medical research at the Ministry of Health and Sports, the republic union of Myanmar, World Health Organization (WHO) ethics committee, the ethics board at Niigata University and the National Institute of Public Health in Japan. Written informed consent was obtained from all participants before the interviews. Voluntary participation and the right to withdraw participation at any time were assured. The study conformed to the principles of the Declaration of Helsinki.

## 3. Results

### 3.1. Characteristics of Participants in Urban and the Rural Areas

Table 1 shows the respondents’ socio-demographic characteristics for both the complete case of Yangon region and Bago region. Although there were no significant differences in two groups with respect to distribution of age, experiences of illness during preceding year and religion, there were significant differences between the groups in the median GDS score and the other variables. Participants who lived in Bago region tend to have high GDS score, more likely to be male and visit religious facilities, with a lower educational attainment and wealth index (Table 1).

### 3.2. Associations with Depressive Symptoms

Depressive symptoms were positively associated with living in rural areas (B = 0.42; 95% confidence interval (CI): 0.12,0.72), female (B = 0.55; 95% CI: 0.31,0.79), illness during the preceding year (B = 0.68; 95% CI: 0.45,0.91) and non-Buddhist religion (B = 0.57; 95% CI: 0.001,1.15) and protectively associated with education to middle school level or higher (B = −0.61; 95% CI: −0.94, −0.28) and the frequency of visits to religious facilities (B = −0.20; 95% CI: −0.30, −0.10; Table 2).

### 3.3. Associations with Depressive Symptoms According to Region

Some of the factors were still positively associated with depressive symptoms in both rural and urban areas in the linear multiple regression analysis. However, they were protectively associated with middle or high wealth index (B = −0.58; 95% CI: −1.08, −0.08) only in the urban area and primary education (B = −0.41; 95% CI: −0.79, −0.02) and frequency of visits to religious facilities (B = −0.34; 95% CI: −0.52, −0.17) only in the rural area (Table 2).

### 3.4. Associations with Depressive Symptoms According to Gender and Region

Gender and regional stratified analyses were performed (Table 3 and Table 4). In men in the urban area, no variables were associated with depressive symptoms. In women in the urban area, depressive symptoms were positively associated with illness during the preceding year (B = 0.78; 95% CI: 0.36,1.20) and protectively associated with education to middle school level or higher (B = −0.67; 95% CI: −1.23, −0.11), middle or high wealth index (B = −0.92; 95% CI: −1.59, −0.25) and the frequency of visiting religious facilities (B = −0.20; 95% CI: −0.38, −0.03) (Table 3). In men in the rural area, depressive symptoms were positively associated only with illness during the preceding year (B = 0.87; 95% CI: 0.33, 1.42). In women in the rural area, depressive symptoms were positively associated with illness during the preceding year (B = 0.83; 95% CI: 0.36, 1.30) and protectively associated with primary education (B = −0.62; 95% CI: −1.12, −0.12) and the frequency of visits to religious facilities (B = −0.44; 95% CI: −0.68, −0.21) (Table 4).

## 4. Discussion

The main contribution of this study was the identification of rural–urban differences in the associations between depressive symptoms and social, health and religious characteristics in older adults of the two rural and urban areas in Myanmar. Overall, older adults in the rural area were more likely to experience depressive symptoms relative to those in the urban area, even after adjusting for potential confounding factors. This result was consistent with previous findings of a regional survey and a meta-analysis conducted in China [28,29,30,31]. In the current study, depressive symptoms were associated with sex differences, physical health, educational attainment and wealth index in both the urban and rural areas. They were associated only with the frequency of visiting religious facilities in the rural are; however, following stratification according to gender, with only female older adults in both the urban and rural areas. There are various possible reasons why older adults in the rural area were more likely to experience depressive symptoms, relative to those in the urban area, in Myanmar.

Myanmar is classified as a country with a critical shortage of health workers, which jeopardizes access to health services, resulting in poorer health status in people in hard-to-reach, or rural, areas [51]. Healthcare disadvantages could be particularly detrimental to older adults’ mental health, as their need for healthcare is greater relative to that of younger people [29].

A previous study suggested that behavioral risk factors for noncommunicable diseases, such as smoking daily, low fruit and vegetable consumption and low levels of physical activity, in rural residents were higher relative to those observed in urban residents in Myanmar [37]. This could lead to physical illness, which is known to increase the risk of depression in older adults [52]. Although we adjusted for illness during the preceding year, health disparities could increase depressive symptoms in older adults in the rural area in this study.

People with low socioeconomic status are also vulnerable to the development of depressive symptoms [29,53,54] and substantial disparities in housing and living conditions exist between the rural and urban populations of Myanmar, as in China [29]. According to the population census, which has been conducted for three decades in Myanmar, in 2014, 77.5% and 15% of the urban and rural populations, respectively, reported electricity as the source of household lighting. In addition, 52% of households in urban areas and 92% of households in rural areas use firewood or charcoal for cooking [55]. The Poverty Report illustrated that poverty was strongly correlated with residential area: rural inhabitants were 2.7 times more likely than urban inhabitants to be poor in Myanmar [56]. In the current study, almost 70% of older adults in rural areas showed low wealth index scores and almost 50% had low educational level; these proportions were significantly higher relative to those observed in the urban area. Even after adjusting for these variables, living in the rural area was a significant predictor of depressive symptoms. Following stratification according to region (urban vs. rural areas), however, wealth index was associated with depressive symptoms only in the urban area. Inequalities in urban areas have increased with the growing incidence of homelessness, unemployment, social deprivation and health problems [57]. This could have affected depressive symptoms in older adults living in the urban area.

Previous studies suggested that religiousness tended to be experienced and expressed more strongly by women [58,59,60], older participants [58,59,60] and residents of rural areas [60,61]. These findings were consistent with the result indicating that the frequency of visiting religious facilities was protectively associated with depressive symptoms in the rural area. Despite ongoing changes in rural Myanmar, a considerable Rural–urban gap in access to material (land, farms and savings), human resources (education, knowledge and skills) and social resources (trust-based bonds) could affect residents’ well-being and the psychological process of coping with diseases including depression [61]. The findings could also be explained by issues of gender inequality in Myanmar. According to a report published by the Asian Development Bank, older women living alone were among the poorest in the population and many women in male-headed households were considered poor because they were powerless and suffered serious deprivation in all areas of capability in Myanmar. In fact, some of the poorest women lived in male-headed wealthier households, particularly if they were of an older age [62]. As older women tend to seek salvation from religiousness, the frequency of visiting religious facilities could have been protectively associated with depressive symptoms in the current study; however, it should be noted that religiousness and the frequency of visits to religious facilities do not necessarily have the same meaning.

The findings of this study should be interpreted within the context of several limitations. Since the response rate of more than 85% in both the Yangon and Bago regions, was quite high by survey standards [63], our findings may be representative in the two regions. However, it is unknown whether these findings are generalizable beyond the Yangon and Bago regions of Myanmar. Myanmar is composed of seven regions and seven states. People who live in the Bago region could enjoy greater access to urban areas and be more likely to access to health facilities relative to those living in other rural areas of the country. Therefore, it is difficult to generalize the study findings to the population in areas where access to urban areas and health services are limited. However, we can estimate situations of older adults in other regions by the level of urbanization of the selected regions. Ideally, this social epidemiological survey should be extended to include the surrounding regions and states of the entire country [64]. In addition, the self-administered nature of the questionnaire did not allow objective assessment of participants’ situations and our measurement of depressive symptoms was based only on GDS scores, without corroborating clinical evaluation. The assessment based on self-report may have caused social desirability bias, resulting in misreporting of depressive symptoms. However, the GDS is a well validated instrument for assessing depressive symptoms and used widely in international research [43,45,46,47,51,65]. Furthermore, reverse causality could have occurred because of the nature of the cross-sectional design and further longitudinal studies are required to resolve this issue. Despite these limitations, the study identified critical and potential factors affecting depressive symptoms in older adults of the two regions in Myanmar.

## 5. Conclusions

Religion and wealth could have different levels of association with depression between older adults in the urban and rural areas and men and women. Intervention programs for depression in older adults should consider regional and sex differences in the roles of religion and economic disparities in rural and urban areas of Myanmar.

## Figures and Tables

**Table 1 ijerph-18-02818-t001:** Socio-demographic characteristics.

		Total		Yangon		Bago		
		*n* = 1200	%	*n* = 600	%	*n* = 600	%	*p*-Value
Median GDS score	(smallest-largest)	2	(0–10)	2	(0–9)	3	(0–10)	<0.001 ^a^
Age	60–69	670	55.8%	351	58.5%	319	53.2%	0.14 ^b^
	70–79	380	31.7%	175	29.2%	205	34.2%	
	80+	150	12.5%	74	12.3%	76	12.7%	
Sex	Male	483	40.3%	222	37.0%	261	43.5%	0.02 ^b^
	Female	717	59.8%	378	63.0%	339	56.5%	
Illness during preceding year	No	582	48.5%	285	47.5%	297	49.5%	0.54 ^b^
	Yes	615	51.3%	312	52.0%	303	50.5%	
	Missing	3	0.3%	3	0.5%	0	0.0%	
Education	No school/Monastic	396	33.0%	127	21.2%	269	44.8%	<0.001 ^b^
	Some/Finished primary	417	34.8%	163	27.2%	254	42.3%	
	Middle school or higher	387	32.3%	310	51.7%	77	12.8%	
Wealth index	Low	480	40.0%	64	10.7%	416	69.3%	<0.001 ^b^
	Middle/High	718	59.8%	535	89.2%	183	30.5%	
	Missing	2	0.2%	1	0.2%	1	0.2%	
Religion	Buddhism	1147	95.6%	569	94.8%	578	96.3%	0.21 ^b^
	Other	53	4.4%	31	5.2%	22	3.7%	
Frequency of religious visits								
	None	103	8.6%	79	13.2%	24	4.0%	<0.001 ^b^
	A few times per year	234	19.5%	124	20.7%	110	18.3%	
	1–3 times per month	280	23.3%	128	21.3%	152	25.3%	
	Once per week	491	40.9%	217	36.2%	274	45.7%	
	2–3 times per week	50	4.2%	20	3.3%	30	5.0%	
	≥4 times per week	42	3.5%	32	5.3%	10	1.7%	

^a^: Wilcoxon rank-sum test, ^b^: chi-square test. GDS = Geriatric Depression Scale. Since GDS did not approximate a normal distribution, Wilcoxon rank-sum test was adopted.

**Table 2 ijerph-18-02818-t002:** Multivariate adjusted association between sociodemographic factors and Geriatric Depression Scale scores among older adults of the two regions in Myanmar.

	All				Yangon				Bago			
	*n* = 1182	B	95%CI		*n* = 591	B	95%CI		*n* = 591	B	95%CI	
**Region**	Yangon											
	Bago	0.42	0.12	0.72								
**Age**	60–69											
	70–79	−0.13	−0.39	0.13		0.02	−0.33	0.37		−0.22	−0.61	0.17
	80+	0.22	−0.36	0.39		0.03	−0.47	0.54		0.03	−0.54	0.59
**Sex**	Male											
	Female	0.55	0.31	0.79		0.64	0.31	0.97		0.42	0.06	0.78
**Illness during preceding year**	No											
	Yes	0.68	0.45	0.91		0.52	0.22	0.83		0.86	0.51	1.22
**Education**	No school/Monastic											
	Some/Finished primary	−0.28	−0.57	0.01		−0.09	−0.54	0.36		−0.41	−0.79	−0.02
	Middle school or higher	−0.61	−0.94	−0.28		−0.47	−0.89	−0.05		−0.69	−1.28	−0.11
**Wealth index**	Low											
	Middle/High	−0.22	−0.52	0.08		−0.58	−1.08	−0.08		−0.06	−0.44	0.33
**Religion**	Buddhism											
	Other	0.57	0.001	1.15		0.65	−0.05	1.35		0.50	−0.44	1.45
**Frequency of religious visits**		−0.20	−0.30	−0.10		−0.12	−0.24	0.00		−0.34	−0.52	−0.17

Adj R-squared = 0.095; Adj R-squared = 0.072; Adj R-squared = 0.083; B: Unstandardized regression coefficient, CI: Confidence Interval. Linear regression analyses were performed.

**Table 3 ijerph-18-02818-t003:** Multivariate adjusted association between sociodemographic factors and Geriatric Depression Scale scores among the urban male & female older adults.

	Yangon Male				Yangon Female			
	*n* = 220	B	95%CI		*n* = 371	B	95%CI	
**Age**	60–69							
	70–79	0.41	−0.09	0.91		−0.19	−0.66	0.29
	80+	0.56	−0.10	1.21		−0.24	−0.95	0.47
**Illness during preceding year**	No							
	Yes	0.06	−0.36	0.48		0.78	0.36	1.20
**Education**	No school/Monastic							
	Some/Finished primary	0.69	−0.10	1.47		−0.29	−0.85	0.27
	Middle school or higher	0.03	−0.59	0.65		−0.67	−1.23	−0.11
**Wealth index**	Low							
	Middle/High	0.11	−0.63	0.84		−0.92	−1.59	−0.25
**Religion**	Buddhism							
	Other	0.15	−1.04	1.34		0.81	−0.06	1.68
**Frequency of religious visits**		−0.05	−0.20	0.10		−0.20	−0.38	−0.03

Adj R-squared = 0.007; Adj R-squared = 0.064; B: Unstandardized regression coefficient, CI: Confidence Interval. Linear regression analyses were performed.

**Table 4 ijerph-18-02818-t004:** Multivariate adjusted association between sociodemographic factors and Geriatric Depression Scale scores among the rural male & female older adults.

	Bago Male				Bago Female			
	*n* = 259	B	95%CI		*n* = 332	B	95%CI	
**Age**	60–69							
	70–79	−0.28	−0.88	0.32		−0.23	−0.75	0.30
	80+	0.01	−0.93	0.96		−0.06	−0.78	0.65
**Illness during preceding year**	No							
	Yes	0.87	0.33	1.42		0.83	0.36	1.30
**Education**	No school/Monastic							
	Some/Finished primary	−0.17	−0.79	0.45		−0.62	−1.12	−0.12
	Middle school or higher	−0.62	−1.40	0.15		−0.94	−1.93	0.04
**Wealth index**	Low							
	Middle/High	−0.19	−0.78	0.40		0.03	−0.50	0.55
**Religion**	Buddhism							
	Other	−0.36	−1.86	1.14		1.25	−0.02	2.52
**Frequency of religious visits**		−0.24	−0.51	0.02		−0.44	−0.68	−0.21

Adj R-squared = 0.073; Adj R-squared = 0.074; B: Unstandardized regression coefficient, CI: Confidence Interval. Linear regression analyses were performed.

## Data Availability

An ethics committee has placed an ethical restriction on sharing de-identified data, since the data may contain sensitive information regarding the respondents’ physical and mental health.

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
