# Peer review of "Rural–Urban Differences in the Factors Affecting Depressive Symptoms among Older Adults of Two Regions in Myanmar"

_ijerph, 2021, doi:10.3390/ijerph18062818_

Round 1

Reviewer 1 Report

The manuscript “Rural-urban differences in the factors affecting depressive symptoms among older adults of the two regions in Myanmar” investigated the associations between depressive symptoms and residing in urban or rural locations, gender, wealth, education, and attendance to religious services. Increased depressive symptoms were associated with living in rural areas and having been ill recently, whereas higher education and visit to religious facilities appeared to be protective.

The study is interesting and well designed.

Some suggestions to improve the manuscript:

The sentences at lines 138-141 should be corrected.

Likewise, the sentence starting at line 240 does not flow correctly.

Lines 147-149: please explain the meaning of score intervals to ease the comprehension.

In Table 1, the title Mean (SD) is in a mistaken position and the title GDS is missing.

Instructions to authors were not removed from the template (lines 317 and following, 356 and following, 361 and following).

Author Response

Response to reviewer 1

Re:  IJERPH (ISSN 1660-4601) Manuscript ID ijerph-1071247

Rural-urban differences in the factors affecting depressive symptoms among older adults of the two regions in Myanmar

Response to reviewer 1

The authors really appreciate your thorough consideration and helpful comments regarding our manuscript. We have addressed each of these suggestions in the responses provided below.

Reviewer 1 comments:

Reviewer1-1 comment:

The sentences at lines 138-141 should be corrected.

Our reply 1-1:

Following the comment from the reviewer, we have revised the sentence as follows:

Participants were older adults aged 60 years or older who came to the center's out-patient clinic. We recruited 25 respondents who provided consent to participate in the pilot study, in June 2018.

Reviewer1-2 comment:

Likewise, the sentence starting at line 240 does not flow correctly.

Our reply 1-2:

Following the comment from the reviewer, we have revised the sentence as follows:

Overall, older adults in the rural area were more likely to experience depressive symptoms relative to those in the urban area, even after adjusting for potential confounding factors.

Reviewer1-3 comment:

Lines 147-149: please explain the meaning of score intervals to ease the comprehension.

Our reply 1-3:

The scores range from 0 to 15 points, with a higher score indicating a more depressive state (49).

Reviewer1-4 comment:

In Table 1, the title Mean (SD) is in a mistaken position and the title GDS is missing.

Our reply 1-4:

Following the comment from reviewers 1 & 2 (comment 2-8 & 2-10), we have revised Table 1 as follows.

Table 1. Socio-demographic characteristics

Total

Yangon

Bago

n=1,200

%

n=600

%

n=600

%

p-value

Median GDS score

(smallest-largest)

2

(0-10)

2

(0-9)

3

(0-10)

<.001a

Age

 60-69

670

55.8%

351

58.5%

319

53.2%

0.14b

 70-79

380

31.7%

175

29.2%

205

34.2%

 80+

150

12.5%

74

12.3%

76

12.7%

Sex

Male

483

40.3%

222

37.0%

261

43.5%

0.02b

Female

717

59.8%

378

63.0%

339

56.5%

Illness during preceding year

No

582

48.5%

285

47.5%

297

49.5%

0.54b

Yes

615

51.3%

312

52.0%

303

50.5%

Missing

3

0.3%

3

0.5%

0

0.0%

Education

No school/Monastic

396

33.0%

127

21.2%

269

44.8%

<.001b

Some/Finished primary

417

34.8%

163

27.2%

254

42.3%

Middle school or higher

387

32.3%

310

51.7%

77

12.8%

Wealth index

Low

480

40.0%

64

10.7%

416

69.3%

<.001b

Middle/High

718

59.8%

535

89.2%

183

30.5%

Missing

2

0.2%

1

0.2%

1

0.2%

Religion

Buddhism

1147

95.6%

569

94.8%

578

96.3%

0.21b

Other

53

4.4%

31

5.2%

22

3.7%

Frequency of religious visits

None

103

8.6%

79

13.2%

24

4.0%

<.001b

A few times per year

234

19.5%

124

20.7%

110

18.3%

1-3 times per month

280

23.3%

128

21.3%

152

25.3%

Once per week

491

40.9%

217

36.2%

274

45.7%

2-3 times per week

50

4.2%

20

3.3%

30

5.0%

≥4 times per week

42

3.5%

32

5.3%

10

1.7%

a: Wilcoxon rank-sum test, b: chi-square test.

GDS = Geriatric Depression Scale

*Since GDS did not approximate a normal distribution, Wilcoxon rank-sum test was adopted.

Reviewer1-5 comment:

Instructions to authors were not removed from the template (lines 317 and following, 356 and following, 361 and following).

Our reply1-5

We have removed the temples and added acknowledgement as follows:

We would like to thank all the study participants and express our gratitude to Professor Than Win Nyunt, from the Department of Geriatric Medicine, Yangon General Hospital, Yangon, Myanmar; Infectious Diseases Research Centre of Niigata University members, particularly Professor Reiko Saito, and Professor Hisami Watanabe. In addition, we thank Ms. Saw Thu Nander, Mr. Yi Mynt Kyaw, and Myanmar Perfect Research team members, who were deeply involved in the project implemented to conduct the survey. We wish to express our gratitude to the Japan Gerontological Evaluation Study principle investigator, Professor Katsunori Kondo, and core members, Dr. Naoki Kondo, Dr. Jun Aida, Dr. Toshiyuki Ojima, and Dr. Masashige Saito, who provided helpful advice regarding the project; Dr. Hiroshi Murayama from the University of Tokyo, who advised the authors on the aging survey based on his professional experience; Ms. Akiko Tomita and Ms. Naoko Ito from the Japan International Cooperation Agency, who supported the conduct of the survey; Ms. Tomoko Manabe, who performed excellent secretarial support during the entire study; and Dr. Reiko Hayashi from the National Institute of Population and Social Security Research, Japan, who provided supportive advice.

Reviewer 2 Report

The manuscript entitled “Rural-urban differences in the factors affecting depressive symptoms among older adults of the two regions in Myanmar” presents interesting issue, but some areas must be corrected.

General:

The manuscript is shabbily prepared and should be corrected to be formatted according to the instructions for authors – e.g. references section – ref 1-8, references in the text, etc

Abstract:

Authors should formulate the aim of their study (e.g. “The aim of the study was”), instead of only presenting what was done

Authors should present the applied methodology

Authors should present specific numeric results of their study accompanied by the results of their statistical analysis.

Introduction:

Authors should present comprehensive knowledge about factors influencing depression and general mental health, including e.g. family related factors (https://www.ncbi.nlm.nih.gov/pmc/articles/PMC1352328/), diet that is followed (https://pubmed.ncbi.nlm.nih.gov/31906271/), social capital (https://www.ncbi.nlm.nih.gov/pmc/articles/PMC5834404/)

Materials and Methods:

Authors indicated that their sample may have been not representative, but they should in fact verify it and clearly indicate if it was or not.

The applied questionnaire should be presented in details. Authors should present the questions that were asked, the form of the questions, information about previous validation of the questionnaire, etc.

Authors stated that they verified validity of the applied questionnaire, but they should describe the results of the validation study (including assessment of validity and reproducibility), or clearly indicate which data did they gather

It seems that Authors did not verify the normality of distribution – they should verify it, and indicate which statistical test was used for verification.

For normally distributed data Authors should present mean and SD values, but for the other distributions – present median, min and max values

Authors should apply adequate statistical tests, that are based on the distribution.

Results:

Authors should verify the representativeness of the studied group

It seems that Authors did not verify the normality of distribution – they should verify it, and indicate which statistical test was used for verification.

For normally distributed data Authors should present mean and SD values, but for the other distributions – present median, min and max values

Authors should not reproduce in the text data that are already presented in tables

Tables should be stand-alone ones – be able to be understand without reading the manuscript, so Authors should explain everything needed in footnotes.

Discussion:

Authors should in their discussion include 3 areas: (1) compare gathered data with the results by other authors, (2) formulate implications of the results of their study and studies by other authors, (3) formulate the future areas which should be studied

Conclusions:

Conclusions can not be too general and should include only those aspects which may be indicated based on the conducted study.

Author Response

Response to reviewer 2

Re:  IJERPH (ISSN 1660-4601) Manuscript ID ijerph-1071247

Rural-urban differences in the factors affecting depressive symptoms among older adults of the two regions in Myanmar

The authors really appreciate your thorough consideration and helpful comments regarding our manuscript. We have addressed each of these suggestions in the responses provided below.

Reviewer 2 comments:

Reviewer 2-1 comment

General:

The manuscript is shabbily prepared and should be corrected to be formatted according to the instructions for authors – e.g. references section – ref 1-8, references in the text, etc

Our reply 2-1

We have removed the instructions for authors.

Reviewer 2-2 comment

Abstract:

Authors should formulate the aim of their study (e.g. “The aim of the study was”), instead of only presenting what was done

Our reply 2-2

Following the comment from the reviewer, we have revised the sentence as follows:

The aim of the study was to investigate rural-urban differences in depressive symptoms in terms of the risk factors among older adults of two regions in Myanmar.

Reviewer 2-3 comment

Abstract:

Authors should present the applied methodology

Our reply 2-3

We thank the reviewer for this fruitful suggestion. We afraid to say, we have a strict limitation of word count.

However, we have added the methodology as follows. We hope it is within the allowable range.

This cross-sectional study, conducted between September and December, 2018, used a multistage sampling method to recruit participants from the two regions, for face-to-face interviews.

Reviewer 2-4 comment

Abstract:

Authors should present specific numeric results of their study accompanied by the results of their statistical analysis.

Our reply 2-4

For the same reason as above, we have not shown numeric results. However, we have added them following the comment from the reviewer as follows. We hope it is within the allowable range.

Depressive symptoms were positively associated with living in rural areas (B= 0.42; 95% confidence interval (CI): 0.12,0.72), female (B = 0.55; 95% CI: 0.31,0.79), illness during the preceding year (B = 0.68; 95% CI: 0.45,0.91), and non-Buddhist religion (B = 0.57; 95% CI: 0.001,1.15), and protectively associated with education to middle school level or higher (B = -0.61; 95% CI: -0.94,-0.28), and the frequency of visits to religious facilities (B = -0.20; 95% CI: -0.30,-0.10). In women in urban areas, depressive symptoms were positively associated with illness during the preceding year (B = 0.78; 95% CI: 0.36, 1.21), and protectively associated with education to middle school level or higher (B = -0.67; 95% CI: -1.23,-0.11), middle or high wealth index (B = -0.92; 95% CI: -1.59,-0.25), and the frequency of visits to religious facilities (B = -0.20; 95% CI: -0.38,-0.03). In men in rural areas, illness during the preceding year was positively associated with depressive symptoms (B = 0.87; 95% CI: 0.33, 1.42). In women in rural areas, depressive symptoms were positively associated with illness during the preceding year (B = 0.83; 95% CI: 0.36, 1.30), and protectively associated with primary education (B = -0.62; 95% CI: -1.12,-0.12), and the frequency of visits to religious facilities (B = -0.44; 95% CI: -0.68,-0.21).

Reviewer 2-5 comment

Introduction:

Authors should present comprehensive knowledge about factors influencing depression and general mental health, including e.g. family related factors (https://www.ncbi.nlm.nih.gov/pmc/articles/PMC1352328/), diet that is followed (https://pubmed.ncbi.nlm.nih.gov/31906271/), social capital (https://www.ncbi.nlm.nih.gov/pmc/articles/PMC5834404/)

Our reply 2-5

Following the comment from the reviewer, we have added the followings. Since our target was older adults, we have cited a paper on depression in older adults.

Risk factors leading to the development of late-life depression likely comprise complex interactions among genetic vulnerabilities, cognitive diathesis, age-associated neurobiological changes, and stressful events (7). The common pathways to depression in older adults are assumed to be curtailment of daily activities and changes in social environments such as family functionality, social contact, and eating status (7-10).

Reviewer 2-6 comment

Materials and Methods:

Authors indicated that their sample may have been not representative, but they should in fact verify it and clearly indicate if it was or not.

Results:

Authors should verify the representativeness of the studied group

Our reply 2-6

Following the comment from the reviewer, we have added the followings in the limitation section.

Since the response rate of more than 85% in both the Yangon and Bago regions, was quite high by survey standards (63), our findings may be representative in the two regions. However, it is unknown whether these findings are generalizable beyond the Yangon and Bago regions of Myanmar. Myanmar is composed of seven regions and seven states. People who live in the Bago region could enjoy greater access to urban areas and be more likely to access to health facilities relative to those living in other rural areas of the country. Therefore, it is difficult to generalize the study findings to the population in areas where access to urban areas and health services are limited. However, we can estimate situations of older adults in other regions by the level of urbanization of the selected regions. Ideally, this social epidemiological survey should be extended to include the surrounding regions and states of the entire country (64).

Reviewer 2-7 comment

Materials and Methods:

The applied questionnaire should be presented in details. Authors should present the questions that were asked, the form of the questions, information about previous validation of the questionnaire, etc. Authors stated that they verified validity of the applied questionnaire, but they should describe the results of the validation study (including assessment of validity and reproducibility), or clearly indicate which data did they gather

Our reply 2-7

We have determined that the following procedure has ensured the face validity and content validity of the questionnaire. We have also added the questions for GDS which was the dependent variable in this study as follows. In addition, the applied questionnaire has been added in Appendix 1 & 2. We have followed the “Linguistic Validation Manual for Health Outcome Assessments (36)”.

2.2. Study tools

A structured questionnaire was developed for face-to-face interviews, following the Japan Gerontological Evaluation Study (JAGES), which is a nationwide, population-based, prospective cohort study for older community-dwelling Japanese adults in Japan (39). The linguistic translation and validation process followed the “Linguistic Validation Manual for Health Outcome Assessments (40). It was first translated into English. Thereafter, it was translated into the local language (Burmese) and back translated into English to ensure clarity and consistency.

Research staff from the Myanmar Perfect Research Company, a group that conducts epidemiological surveys in Myanmar, were hired. The interviewers were recruited from the company. Before the commencement of the actual survey, a two-day training course on the research protocol, administration of the questionnaire, and ethical concerns was conducted for the interviewers.

A pilot study was carried out before the actual survey in Urban Health Center, Dagon township, Yangon. Participants were older adults aged 60 years or older who came to the center's out-patient clinic. We recruited 25 respondents who provided consent to participate in the pilot study, in June 2018. During the pilot study, the interviewers ensured sequence, flow, and clarity of the study. After the feedback from the interviewers, the questionnaire was revised accordingly. We have determined that the above procedure ensured the face validity and content validity of the applied questionnaire (Appendix 1 & 2).

The inclusion criteria were age of ≥60 years and residence in a selected ward or village tract. The exclusion criteria were being bed-ridden or having severe dementia. Severe dementia was defined as an Abbreviated Mental Test (AMT) score of ≤6 (41, 42).

We assessed depressive symptoms using the 15-item version of the GDS (GDS-15), which was validated previously (43-47). The GDS involves a simple yes/no format that is easy to administer and score (47, 48). GDS includes the following questions: 1) Are you satisfied with your current life?; 2) Do you sometimes feel there is no point in living?; 3) Do you think your energy for daily life or your interest in what’s going on in the world has been decreasing?; 4) Do you feel your life is empty?; 5) Do you often feel bored?; 6) Do you usually feel good?; 7) Do you feel something bad is going to happen?; 8) Do you think you are fortunate?; 10) Do you prefer staying at home over going out?; 11) Do you think you are more forgetful than others?; 12) Do you think life is wonderful?; 13) Do you feel full of energy?; 14) Do you think there is no hope in your life?; 15) Do you think others are better off than you are? The scores range from 0 to 15 points, with a higher score indicating a more depressive state (49).

Variables with a variance inflation factor (VIF) of > 5 with other variables were excluded. The remaining variables reflecting sociodemographic characteristics were entered in a linear regression model. They included information regarding residential area (Yangon or Bago regions), sex, age, illness during the preceding year, educational level (no school, monastic, some/all primary school, middle/high school or higher), wealth index, religion (Buddhism or other), frequency of visits to religious facilities (none, a few times per year, one to three times per month, once per week, twice or three times per week, four or more times per week). The wealth index used as an economic indicator was calculated from household asset items using a method described in a previous report (50). Principal component analysis was performed on the asset items (e.g., radio, black & white television, color television, Video/DVD player, electric fan, refrigerator, computer, store-bought furniture, personal music player, washing machine, gas cooker, electric cooker or rice cooker, air conditioner, bicycle, motorcycle, van/truck, microwave oven, mobile telephone, and internet), and the principal component score was calculated based on the participants’ possession of each item and used as the wealth index.

Reviewer 2-8 comment

Materials and Methods:

For normally distributed data Authors should present mean and SD values, but for the other distributions – present median, min and max values. Authors should apply adequate statistical tests, that are based on the distribution.

It seems that Authors did not verify the normality of distribution – they should verify it, and indicate which statistical test was used for verification.

Our reply 2-8

Following the comment from the reviewer, we have added the followings in the section of “Statistical analysis”.

We calculated rates for each category of sociodemographic variables for the 1,200 participants. Skewness and kurtosis were calculated for the distribution of GDS scores. Then, Wilcoxon rank-sum test and chi-square tests were used to compare GDS scores, and socio-demographic variables between urban and rural areas.

Reviewer 2-9 comment

Results:

Authors should not reproduce in the text data that are already presented in tables

Our reply 2-9

Following the comment from the reviewer, we have confirmed No. 4 in “9 Tips on presenting your tables effectively (https://www.editage.com/insights/presenting-your-tables-effectively)”. Then, we have carefully checked the results in the manuscript.    

Reviewer 2-10 comment

Results:

Tables should be stand-alone ones – be able to be understand without reading the manuscript, so Authors should explain everything needed in footnotes.

Our reply 2-10

Following the comment from the reviewer, we have added footnotes in all Tables as follows.

Footnotes for Table 1.

a: Wilcoxon rank-sum test, b: chi-square test.

GDS = Geriatric Depression Scale

*Since GDS did not approximate a normal distribution, Wilcoxon rank-sum test was adopted. 

Footnotes for Tables 2, 2a & 2b

B: Unstandardized regression coefficient, CI: Confidence Interval.

*Linear regression analyses were performed.

Reviewer 2-11 comment

Discussion:

Authors should in their discussion include 3 areas: (1) compare gathered data with the results by other authors, (2) formulate implications of the results of their study and studies by other authors, (3) formulate the future areas which should be studied

Our reply 2-11

We have included 3 areas which the reviewer suggested.

(1) & (2)

Overall, older adults in the rural area were more likely to experience depressive symptoms relative to those in the urban area, even after adjusting for potential confounding factors. This result was consistent with previous findings of a regional survey and a meta-analysis conducted in China (28-31).

(1) & (2)

A previous study suggested that behavioral risk factors for noncommunicable diseases, such as smoking daily, low fruit and vegetable consumption, and low levels of physical activity, in rural residents were higher relative to those observed in urban residents in Myanmar (37). This could lead to physical illness, which is known to in-crease the risk of depression in older adults (52). Although we adjusted for illness during the preceding year, health disparities could increase depressive symptoms in older adults in the rural area in this study.

(1) & (2)

People with low socioeconomic status are also vulnerable to the development of depressive symptoms (29, 53, 54), and substantial disparities in housing and living conditions exist between the rural and urban populations of Myanmar, as in China (29). According to the population census, which has been conducted for three decades in Myanmar, in 2014, 77.5% and 15% of the urban and rural populations, respectively, re-ported electricity as the source of household lighting. In addition, 52% of households in urban areas and 92% of households in rural areas use firewood or charcoal for cooking (55). The Poverty Report illustrated that poverty was strongly correlated with residential area: rural inhabitants were 2.7 times more likely than urban inhabitants to be poor in Myanmar (56). In the current study, almost 70 % of older adults in rural areas showed low wealth index scores and almost 50% had low educational levels; these proportions were significantly higher relative to those observed in the urban area. Even after adjusting for these variables, living in the rural area was a significant predictor of depressive symptoms.

(1) & (2)

Previous studies suggested that religiousness tended to be experienced and ex-pressed more strongly by women (58-60), older participants (58-60), and residents of rural areas (60, 61). These findings were consistent with the result indicating that the frequency of visiting religious facilities was protectively associated with depressive symptoms in the rural area.

(1) & (2)

Despite ongoing changes in rural Myanmar, a considerable rural-urban gap in access to material (land, farms, and savings), human resources (education, knowledge, and skills), and social resources (trust-based bonds) could affect residents’ well-being and the psychological process of coping with diseases including depression (61). The findings could also be explained by issues of gender inequality in Myanmar. According to a report published by the Asian Development Bank, older women living alone were among the poorest in the population, and many women in male-headed households were considered poor because they were powerless and suffered serious deprivation in all areas of capability in Myanmar. In fact, some of the poorest women lived in male-headed wealthier households, particularly if they were of an older age (62). As older women tend to seek salvation from religiousness, the frequency of visiting religious facilities could have been protectively associated with depressive symptoms in the current study;

(3)

Ideally, this social epidemiological survey should be extended to include the sur-rounding regions and states of the entire country (64).  

(3)

Furthermore, reverse causality could have occurred because of the nature of the cross-sectional design, and further longitudinal studies are required to resolve this issue.

Reviewer 2-12 comment

Conclusions:

Conclusions cannot be too general and should include only those aspects which may be indicated based on the conducted study.

Our reply 2-12

Following the comment from the reviewer, we have revised the conclusion as follows.

Religion and wealth could have different levels of association with depression between older adults in the urban and rural areas and men and women. Intervention programs for depression in older adults should consider regional and sex differences in the roles of religion, and economic disparities in rural and urban areas of Myanmar.

Round 2

Reviewer 2 Report

The manuscript entitled “Rural-urban differences in the factors affecting depressive symptoms among older adults of the two regions in Myanmar” presents interesting issue, but some areas should be corrected.

Abstract:

Authors should present justification of the conducted study

Authors should present the applied methodology

Materials and Methods:

Authors indicated that their sample may have been not representative, but they should in fact verify it and clearly indicate if it was or not.

Results:

Authors should not reproduce in the text data that are already presented in tables

Author Response

Response to Reviewer 2

Re:  IJERPH (ISSN 1660-4601) Manuscript ID ijerph-1071247

Rural-urban differences in the factors affecting depressive symptoms among older adults of the two regions in Myanmar

Response to reviewer 2

The authors really appreciate your thorough consideration and helpful comments regarding our manuscript. We have addressed each of these suggestions in the responses provided below.

Reviewer comment 2-1

Abstract: Authors should present justification of the conducted study

Our reply 2-1

Following your comment, we have added the justification.

The aim of the study was to investigate rural-urban differences in depressive symptoms in terms of the risk factors among older adults of two regions in Myanmar to provide appropriate intervention for depression depending on local characteristics.

Reviewer comment 2-2

Abstract: Authors should present the applied methodology

Our reply 2-2

Following your comment, we have added the applied methodology. We afraid to say, however, we have a strict limitation of word count as we said before. We hope it is within the allowable range.

This cross-sectional study, conducted between September and December, 2018, used a multistage sampling method to recruit participants from the two regions, for face-to-face interviews. Depressive symptoms were assessed using the 15-item version of the Geriatric Depression Scale (GDS).

Reviewer comment 2-3

Materials and Methods:

Authors indicated that their sample may have been not representative, but they should in fact verify it and clearly indicate if it was or not.

Our reply 2-3

Following the comment from the reviewer before, we have added the following sentences and also added 2 references (63) & (64). I hope that the following is responsive to your inquiry.

Since the response rate of more than 85% in both the Yangon and Bago regions, was quite high by survey standards (63), our findings may be representative in the two regions. However, it is unknown whether these findings are generalizable beyond the Yangon and Bago regions of Myanmar. Myanmar is composed of seven regions and seven states. People who live in the Bago region could enjoy greater access to urban areas and be more likely to access to health facilities relative to those living in other rural areas of the country. Therefore, it is difficult to generalize the study findings to the population in areas where access to urban areas and health services are limited. However, we can estimate situations of older adults in other regions by the level of urbanization of the selected regions. Ideally, this social epidemiological survey should be extended to include the surrounding regions and states of the entire country (64).

  1. Fincham JE. Response rates and responsiveness for surveys, standards, and the Journal. American journal of pharmaceutical education. 2008;72(2):43.
  2. Sasaki Y, Shobugawa Y, Nozaki I, Takagi D, Nagamine Y, Funato M, et al. Association between depressive symptoms and objective/subjective socioeconomic status among older adults of two regions in Myanmar. PLoS One. 2021;16(1):e0245489.

Reviewer comment 2-4

Results:

Authors should not reproduce in the text data that are already presented in tables

Our reply 2-4

Following the comment from the reviewer, we have revised the results as follows. As we mentioned before, we have confirmed No. 4 in “9 Tips on presenting your tables effectively (https://www.editage.com/insights/presenting-your-tables-effectively)”. Then, we have carefully checked the results in the manuscript.

Table 1 shows the respondents’ socio-demographic characteristics for both the complete case of Yangon region and Bago region. Although there were no significant differences in two groups with respect to distribution of age, experiences of illness during preceding year, and religion, there were significant differences between the groups in the median GDS score, and the other variables. Participants who lived in Bago region tend to have high GDS score, more likely to be male and visit religious facilities, with a lower educational attainment and wealth index (Table 1).
